# Microwave versus Conventional Sintering of NiTi Alloys Processed by Mechanical Alloying

**DOI:** 10.3390/ma15165506

**Published:** 2022-08-11

**Authors:** Rodolfo da Silva Teixeira, Rebeca Vieira de Oliveira, Patrícia Freitas Rodrigues, João Mascarenhas, Filipe Carlos Figueiredo Pereira Neves, Andersan dos Santos Paula

**Affiliations:** 1Departamento de Engenharia de Materiais, Escola de Engenharia de Lorena, Universidade de São Paulo, Polo Urbo Industrial, Gleba AI-6, Lorena 12602-810, Brazil; 2Seção de Engenharia de Materiais, Instituto Militar de Engenharia (IME), Praça General Tibúrcio 80, Urca, Rio de Janeiro 22290-270, Brazil; 3University of Coimbra, Department of Mechanical Engineering, CEMMPRE, R. Luís Reis Santos, 3030-790 Coimbra, Portugal; 4Laboratório Nacional de Energia e Geologia (LNEG), Estrada do Paço do Lumiar, 22, 1649-038 Lisboa, Portugal

**Keywords:** shape memory alloys, mechanical alloying and milling, microwave processing, mechanical properties, powder metallurgy

## Abstract

The present study shows a comparison between two sintering processes, microwave and conventional sintering, for the manufacture of NiTi porous specimens starting from powder mixtures of nickel and titanium hydrogenation–dehydrogenation (HDH) milled by mechanical alloying for a short time (25 min). The samples were sintered at 850 °C for 15 min and 120 min, respectively. Both samples exhibited porosity, and the pore size results are within the range of the human bone. The NiTi intermetallic compound (B2, R-phase, and B19′) was detected in both sintered samples through X-ray diffraction (XRD) and electron backscattering diffraction (EBSD) on scanning electron microscopic (SEM). Two-step phase transformation occurred in both sintering processes with cooling and heating, the latter occurring with an overlap of the peaks, according to the differential scanning calorimetry (DSC) results. From scanning electron microscopy/electron backscatter diffraction, the R-phase and B2/B19′ were detected in microwave and conventional sintering, respectively. The instrumented ultramicrohardness results show the highest elastic work values for the conventionally sintered sample. It was observed throughout this investigation that using mechanical alloying (MA) powders enabled, in both sintering processes, good results, such as intermetallic formation and densification in the range for biomedical applications.

## 1. Introduction

Technological advancements push research in the search for new materials and properties. So-called “intelligent” or “smart” materials are of great industrial interest because of their wide application due to their properties. In the class of intelligent materials, NiTi alloys stand out for their properties, of which their shape memory effect (SME) and superelasticity (SE) confer a singular value for these alloys [1,2]. NiTi alloys are most widely applied in the aerospace and automotive industries and in the manufacture of biomedical equipment and microdevices [3,4,5,6]. These alloys exhibit three phases in the matrix: the B2 austenite phase, the B19′ martensite phase, and the intermediate premartensitic R-phase. The austenitic phase has a polygonal/equiaxed grain microstructural appearance. In contrast, the martensitic phases, B19′ and R-phase, show a triangular morphology (composed of primary and secondary plates) and “herring-bone” appearance, respectively, where several martensitic phase nuclei originate and grow within each austenitic phase grain [7]. The phases present at room temperature depend on the composition of the alloy. The phase transformations may occur from B2
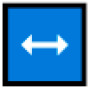
B19′ or with the presence of the intermediary R-phase (B2
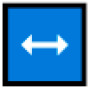
R
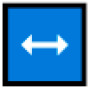
B19′) [7,8].

Bulk NiTi alloy production can occur via different routes that can be divided into two large groups: melting and powder metallurgy. Contamination issues during the melting processes can arise because of titanium’s high reactivity with oxygen and carbon, resulting in the formation of oxides (Ti_4_Ni_2_O) and carbides (TiC) in the liquid state, and the formation of oxides (Ti_2_O) in the solid state. This can be considered a relevant problem because additional phases change the functional properties of NiTi alloys [9,10].

In the case of the powder metallurgy route, there are many advantages compared to the melting processes, namely the possibility of large-scale association within dimension control, cost reduction, impurity reduction, and the feasibility of achieving mechanical properties in ranges of values such as those resulting from conventional metallurgical processes [11]. Mechanical alloying (MA) is an interesting powder metallurgy technique, involving repeated welding, fracturing, and rewelding of the powder particles. This technique allows homogeneous materials from the mixture of two or more elements or compounds [11,12,13]. Conventional sintering (CS) is the simplest form of sintering NiTi alloys. It is relatively inexpensive but usually requires a very long time. However, the main disadvantages of this method are associated with the formation of oxides and other secondary phases in the solid state, which alters the stoichiometric composition of the alloy, affecting the phase transformation temperature [7,14]. Microwave sintering (MW) is a recent method used for the densification of powders, resulting in bulk samples. Its advantages include high heating rates coupled with a low sintering time and reduced energy consumption compared to conventional sintering [15,16,17,18]. The synthesis of porous NiTi alloys is relevant for the manufacture of biomaterials for bone-replacement implants due to their properties, including low density, tissue in-growth ability (securing a firm fixation to the implants), and SME and SE [3,15,16]. It is complicated to evaluate the mechanical behavior of NiTi shape memory alloys (SMAs) due to their asymmetric transformation behavior [19,20]. Usually, the SME and SE are evaluated by conventional tests, for example, the uniaxial tensile, point bending, and compression tests [19,20,21,22]. However, asymmetric behavior is influenced by the grain size (nanoscale, microscale, and macroscale) and the nature of the stress–strain applied [19,20]. The authors of [23] propose that martensitic variants (MVs) are directly associated with asymmetric behavior.

The prime novelty of this research is the short milling time (25 min) in the mechanical alloying process of NiTi alloys, which provided the formation of a lamellar structure. The formation of this lamellar structure in such a short time has never been reported in the open literature for NiTi alloys processed by mechanical alloying. Additionally, in the two sintering processes subsequently evaluated, microwave and conventional sintering, the structure made the formation of a NiTi matrix possible, which was evaluated for its densification, microstructural effect, and micromechanical behavior under ultramicrohardness.

## 2. Materials and Methods

The raw materials used in this study were as follows: The nickel powder (median particle size ≈ 2 µm), supplied by JB Química (São Paulo/SP, Brazil), was obtained by oxi-reduction and had a near-spherical morphology. The titanium powder (particle size < 150.0 µm), donated by BRATS (São Paulo/SP, Brazil), was produced by the hydrogenation–dehydrogenation process (HDH) and had an irregular morphology. Figure 1 shows the schematic diagram of the process adopted in the present study.

The powders were mixed to obtain equiatomic NiTi single-phase alloy and milled in a planetary ball mill PM400 (Retsch, Haan, Germany) at 300 rpm for 25 min (after each 5 min of milling, the process was interrupted for 3 min to cool the vials). These experiments were carried out under an argon atmosphere at room temperature and using stainless steel vials (250 mL) and balls (balls with a diameter of 15 mm). The powder milling charge was 18 g, and the ball/powder weight ratio was 20:1.

After milling, the samples were cold-pressed into green pellets (∅ 8 mm × 14 mm), using a uniaxial pressure of 53 MPa for 60 s, and then vacuum encapsulated in a quartz tube for the realization of the sintering. Microwave (MW) sintering was performed at a target temperature of 850 °C for 15 min and a rate of 50 °C/min in an MW oven from Microwave Research & Applications, Inc. (Carol Stream, IL, USA), operating at 2.45 GHz with a power of 1 kW. The temperature was measured by a platinum-shielded S-type thermocouple. Conventional sintering (CS) was performed in a muffle furnace, a Quimis model, for 120 min at the same temperature and heating rate. The conventional sintered samples were then quenched in water at room temperature.

The specimen’s density and porosity were measured by the Archimedes method, based on standard ABNT NBR 16661:2017. The pore size examinations were carried out using optical microscopy (Olympus, Tokyo, Japan). The average porosity was measured using five optical micrographs taken from the cross-sections of the sintered samples.

Differential thermal analysis (DTA) was performed on the MA-processed powders for 25 min (SETARAM, Provence-Alpes-Cote d′Azur, France). The heating/cooling cycles comprised a temperature range from 22 to 1050 °C (heating/cooling rates of 10 K·min^−1^) under an argon atmosphere.

Differential scanning calorimetry (DSC) was carried out under a DSC-60 (Shimadzu, Kyoto, Japan) fitted with a LN_2_ cooling attachment and inert gas (N_2_) atmosphere. The heating/cooling cycles had temperatures of 150 °C and −150 °C (heating/cooling rate of 10 K·min^−1^), and the phase transformation temperatures extracted from the DSC curves were obtained by the tangent method.

X-ray diffraction (XRD) patterns were recorded in a diffractometer (PANalytical, Worcestershire, UK) with a cobalt anode, operated at 40 kV and 40 mA. Scanning electron microscopy (SEM) analysis was performed using a scanning electronic microscope (SEM) (FEI, Hillsboro, OR, USA) with secondary electron (ETD), backscattered electron (BSE), and electron backscattered diffraction (EBSD) detectors.

Instrumented ultramicrohardness analysis was conducted with a Berkovich indenter (Shimadzu DUH-211S, Kyoto, Japan) using the load–unload cycle method and two different maximum load levels: 1.0 gf/9.81 mN and 20.0 gf/196.1 mN. In each sample, ten tests were performed from the longitudinal cross-section of the sintered samples along the ½ diameter.

The XRD, SEM/EBSD, and instrumented ultramicrohardness specimens, after being cut by a precision cutting machine, were prepared by conventional mechanical grinding followed by electrochemical polishing at room temperature using an electrolyte of H_2_SO_4_ (20%) and CH_3_OH (80%) at 30 V. To reveal the phases, the samples were immersed for 60 s in the following etching solution: 50 mL glycerin (C_3_H_8_O_3_), 18 mL concentrated acetic acid (C_2_H_4_O_2_), 20 mL concentrated nitric acid (HNO_3_), and 16 mL concentrated hydrofluoric acid (HF).

## 3. Results

### 3.1. Mechanical Alloying

The starting morphology of the powders is shown in Figure 2. After the MA process, a lamellar structure was formed, as shown in Figure 2c. The presence of this structure can promote increases in the diffusion processes [24,25,26]. This was a remarkable result; lamellar structure has only been observed in the literature for milling times ranging from 4 to 180 h [27,28,29]. However, even with long times, it is not guaranteed to obtain a lamellar structure because there are other process parameters that must be controlled, as has been observed in [24,28,30].

Figure 2d presents the XRD patterns of the starting powders and MA-processed powders. This analysis reveals that no intermetallic phase was detected and only peaks from the nickel and titanium initial powders were recorded. It is worth mentioning that XRD has a detection limit that depends on the equipment [31,32,33], meaning that if any intermetallic phase is formed below the detection limit, it will not be significant in the face of the detected elements.

The DTA curve is shown in Figure 2e, where the effect of the MA-process was analyzed. The curve shows only one peak, corresponding to the reactions between the nickel and titanium to the formation of the intermetallic compound of the Ni-Ti system. As each powder particle contained nickel and titanium in the form of a lamellar structure, the reaction was slow and had low energy. As a consequence, the DTA peak has low intensity and is very broad [34]. The temperature of the peak (T_p_) was determined to be 468.4 °C (Figure 2e), it should be noted that in a mixture as it is (without MA), this reaction occurs at a higher temperature (±950 °C) and with much higher intensity [35]. This temperature differs significantly from the previous results reported in the literature, as it is lower than those obtained for milling times of 4–20 h [29,35,36].

### 3.2. Sintering Characterization

Table 1 summarizes the density, porosity, and pore size values obtained for the MW and CS samples. For comparison purposes, Table 1 also shows some results found in the literature [16,37].

According to the literature, human bone shows a density of 1.8–2.1 g/cm^3^, the ideal porosity is in the range of 30–90% and the optimal pore size is between 100 µm and 500 µm [16,38,39]. Thus, the MW and CS samples prepared in the present study exhibited porosity and pore size values within the range of the human bone but higher densities. However, the values obtained for the CS sample are higher than the ones obtained for the MW sample.

The MW sample showed values close to those found in the literature for porous NiTi alloys also prepared by MW sintering [16,37] (Table 1). However, ref. [16] used a pore size controller (sieved pure NH_4_HCO_3_) and tested different mesh pore sizes. In that study, the porosity results were similar for all samples independently of the initial mesh of the pore size controller. The pore size values obtained in the present study are higher than the ones obtained for the sample sieved to 200 mesh by [16] but smaller than the others.

The authors of [37] used different concentrations of NH_4_HCO_3_ while keeping the mesh size at 250 (Table 1). When comparing the MW sample sintered in the present study, similar density and pore size values were observed for the sample with 20 wt.% NH_4_HCO_3_ (Table 1). However, when compared to the sample without pore size controllers (0 wt.%) of reference [16], it is seen that the density is smaller (3.31 ± 0.02 g/cm^3^ vs. 5.1 ± 0.20 g/cm^3^) while the porosity (51.3 ± 0.29% vs. 22 ± 0.31%) and the pore size (120 ± 13.84 µm vs. 26 ± 2.76 µm) are higher. It is evident that the MA processing promoted alterations in these properties. The formation of the lamellar structure (Figure 2c) provided the acceleration of the diffusion process during sintering and more uniform porous specimens, as has been previously investigated in literature [40,41].

The DSC curves are shown in Figure 3. The dashed line represents the phase present at room temperature (20.0 °C). In the DSC curve of the MW sample (Figure 3a), during heating, a two-stage martensitic transformation (B2 → R → B19′) was observed. During the cooling, as has been noted, an overlap of the DSC peaks was observed with two-step transformations (B2 → R → B19′) [7,13,42,43,44,45,46]. Regarding the CS sample of the DSC curve, (Figure 3b), equal behavior was also noted.

The transformation temperatures and enthalpy of the MW and CS samples are listed in Table 2. The R_s_ and A_f_ temperatures were above room temperature (20 °C) in both sintering processes. During the cooling and heating, a significant variation in the transformation temperatures was not identified, as they did not surpass one degree Celsius during both sintering processes.

Few studies have observed similar behavior. For example, in [37], two-step martensitic transformations in microwave sintering were identified, and compared to this study, the variations in the R_s_ and A_f_ temperatures were approximately 8 °C lower. However, ref. [47] observed a multi-step transformation during the heating and a two-step transformation during the cooling, in the temperature range of 850 to 1000 °C/1 h (variation of 50 °C). Those authors mixed powders by MA for 3 h.

An increase in enthalpy in the CS sample was observed when compared to the MW sample at both peaks of the cooling and heating curves. As is known, the enthalpy changes as a function of the nickel atomic content [48], and the presence of TiNi_3_ in the MW sample is the reason for the low values of enthalpy.

The XRD patterns of the MW and CS samples are shown in Figure 4. The NiTi phases (B2, R-phase, and B19′) were identified in both samples (Figure 4a), although the R-phase in the CS sample only had one peak with overlap for B2 (1 0 0). In addition to the NiTi phases, TiNi_3_ was also recorded in the MW sample (Figure 4a), and Ti_2_Ni and Ti_3_Ni_4_ in the CS sample (Figure 4b). The XRD results show that the MW sample had more peaks for the R-phase and B19′. In contrast, the CS sample had more for B2 and B19′. These results are not exactly what was expected from the DSC results.

Nevertheless, these results are an improvement when compared to those of other studies related to the MW sintering of NiTi alloys. For example, in reference [15], the formation of NiTi intermetallic phases was not observed after MW sintering at 850 °C and 950 °C of the cold-pressed powders blended in a rotating mixer. The justification for such behavior can possibly be associated with the MA process that, as mentioned previously, led to the formation of a lamellar structure, (Figure 2c), which can promote an increase in the diffusion processes. In the CS sample, in addition to the NiTi phase, only Ti_2_Ni and precipitated Ti_3_Ni_4_ were recorded as secondary phases. This result could be associated with the high sintering time of CS (120 min vs. 15 min).

MA enabled the modification of the diffusion process that occurs in the MW and CS processes by promoting the formation of the B2, R-phase, and B19′ phases and by avoiding the formation of a second phase in the CS sample (TiNi_3_) and MW sample (Ti_2_Ni and Ti_4_Ni_3_). In the literature, it is reported that intermetallic TiNi_3_ and Ti_2_Ni form when pure nickel and titanium powder mixtures are submitted to conventionally sintered samples [7,10].

SEM micrographs of the MW and CS sintered samples before etching are shown in Figure 5a,b. It can be observed that the MW sample (Figure 5a) shows more uniform pores compared to the CS sample (Figure 5b). Considering the short milling time and the relatively low sintering temperature, the obtained results are truly relevant compared to those in the literature [15,47].

After metallographic etching, the previous B2 boundaries were identified in both sintered samples (Figure 5c–f), as highlighted in the green squares. In a single B2 grain, a significant amount of the B19′ and R-phase could be formed, preserving the previous B2 boundary [7,49,50,51,52,53]. In addition, the B19′ was also identified in the CS sample (Figure 5f). These results corroborate the XRD analyses (Figure 4).

The EBSD results are shown in Figure 6. In the image quality (IQ) map, a different microstructure between the MW (Figure 6a) and CS samples (Figure 6b) can be observed. Nevertheless, for the MW sample (Figure 6a), there is a possibility of a signal mixture of the R-phase variants multiple grains formed within a given austenite grain. The phase map results show that the MW sample (Figure 6c) had a higher amount of R-phase and the CS sample (Figure 6d) had a higher amount of B2 and B19′. These results are consistent with those observed in the XRD analysis.

The force vs. depth curves (loading and unloading) of the MW and CS sintered samples are shown in Figure 7. Generically, the curves exhibit heterogeneous behavior. However, one curve of the MW sample, highlighted in Figure 7a, displays a significant elastic return (until 0.02 μm), which indicate a possible SE effect. This result is supported by the XRD (Figure 4) and SEM (Figure 5) analyses, which showed B2 presence. However, the non-complete return possibly indicates that the pressure exerted by the load maximum reached the plastic deformation region by screw dislocations, as noted in the study of [54], who proved stress-induced martensitic transformation during a nanoindentation test of NiTi alloys (superelastic). Nevertheless, each of these measures mechanically requests a small number of grains for the indentation and deformed volume around it, and with this, each curve can translate the mechanical behavior in the function of the crystallographic orientation of these grains [54,55]. In addition, the nature of the strain–stress applied has influence on the asymmetric transformation behavior [19,20].

The results of the total work, elastic work, and plastic work, shown in Figure 8, indicate that the higher average and maximum force values were obtained for the CS sample. This is consistent with what was expected from the XRD and EBSD analyses, in which a B2 majority was recorded.

## 4. Conclusions

The comparative sintering study starting from MA powders milled for 25 min allowed us to conclude that:-The density value obtained for the MW sintered sample (3.31 ± 0.02 g/cm^3^) was closer to human bone (1.8–2.1 g/cm^3^) than the density value obtained for the CS (3.76 ± 0.06 g/cm^3^). The results of porosity and pore size for both processes exhibited results within the range of human bone;-When compared to CS, the overall results obtained from combining MA and MW sintering were very promising, as it allowed for the formation of intermetallic NiTi (B2, R-phase, and B19′) in a very short time and the formation of homogeneous pores;-Two-step transformations were detected on the heating and cooling curves for both sintering processes, with an overlapping peak on the cooling curves, probably due to the presence of a second phase and the intermediate premartensitic R-phase. No significant differences in the transformation temperatures were detected between the sintering methods;-The EBSD results show that the MW sample had a higher amount of R-phase and the CS sample had higher amounts of B2 and B19′;-Through instrumented ultramicrohardness analysis, heterogeneous behavior could be identified based on the curves of both samples, as expected. In MW’s curves, one curve was observed with a significant elastic return. The CS sample showed the highest values of elastic work compared to the MW sample at both maximum forces.

## Figures and Tables

**Figure 1 materials-15-05506-f001:**
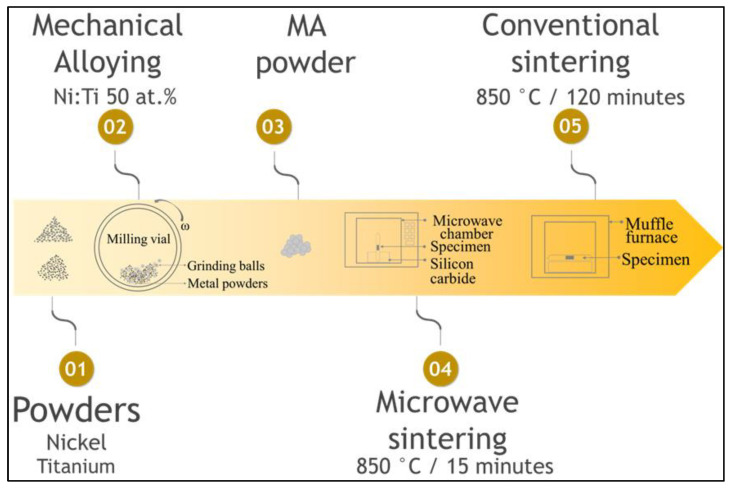
Schematic of the adopted mechanical alloying and sintering processes.

**Figure 2 materials-15-05506-f002:**
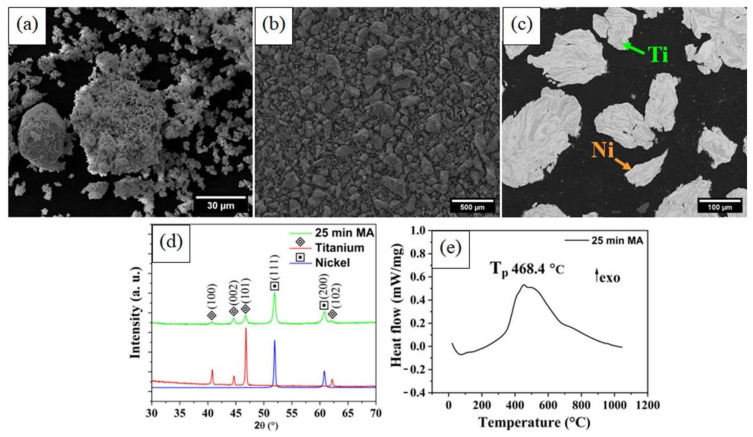
SEM morphology of the elementary powders: (**a**) nickel and (**b**) titanium. (**c**) Micrography of the MA-processed powders. (**d**) XRD patterns of the starting powders and the MA-processed powders. (**e**) DTA heating curve of MA-processed powders.

**Figure 3 materials-15-05506-f003:**
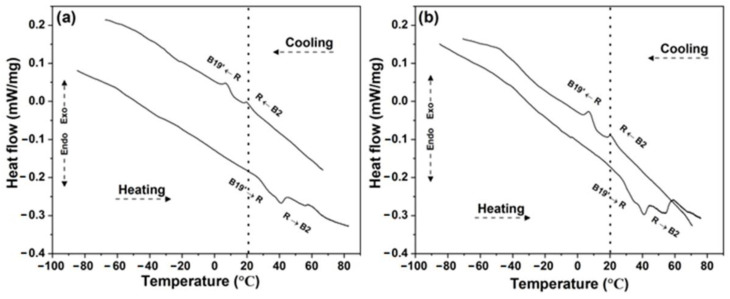
DSC curves showing the phase transformations for (**a**) MW and (**b**) CS samples. Dot line: room temperature (20 °C).

**Figure 4 materials-15-05506-f004:**
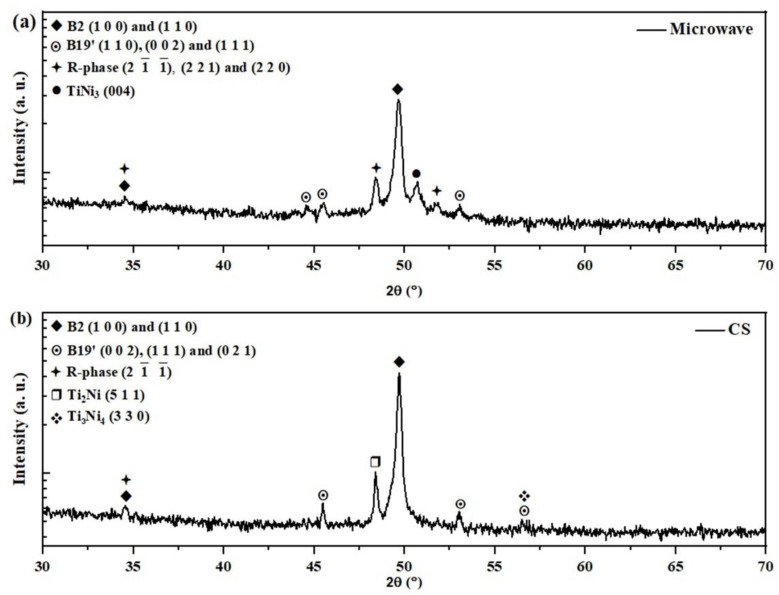
XRD patterns (**a**) MW and (**b**) CS sample.

**Figure 5 materials-15-05506-f005:**
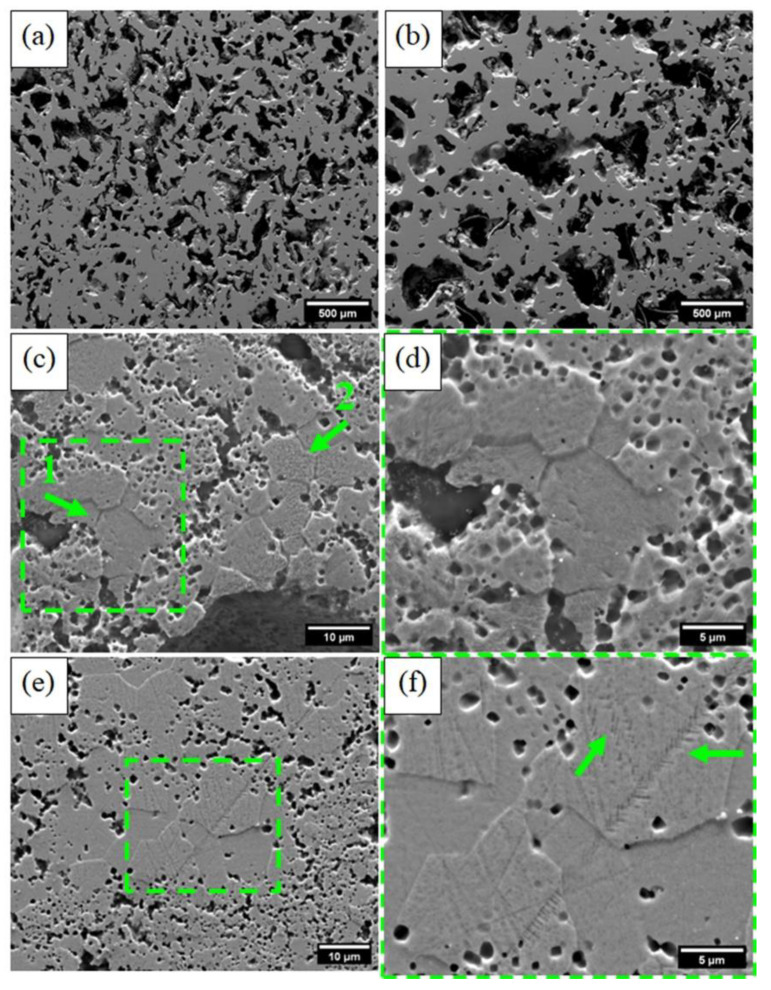
SEM micrographs of the sample porous NiTi: (**a**) MW sample and (**b**) CS sample before etching. (**c**–**f**) After etching to reveal the possible phases. MW sample indicated by arrows 1 and 2: (**c**) general aspect and (**d**) magnification from arrow 1; CS sample: (**e**) general aspect and (**f**) magnification from the square, showing the martensitic phase, indicated by the arrows.

**Figure 6 materials-15-05506-f006:**
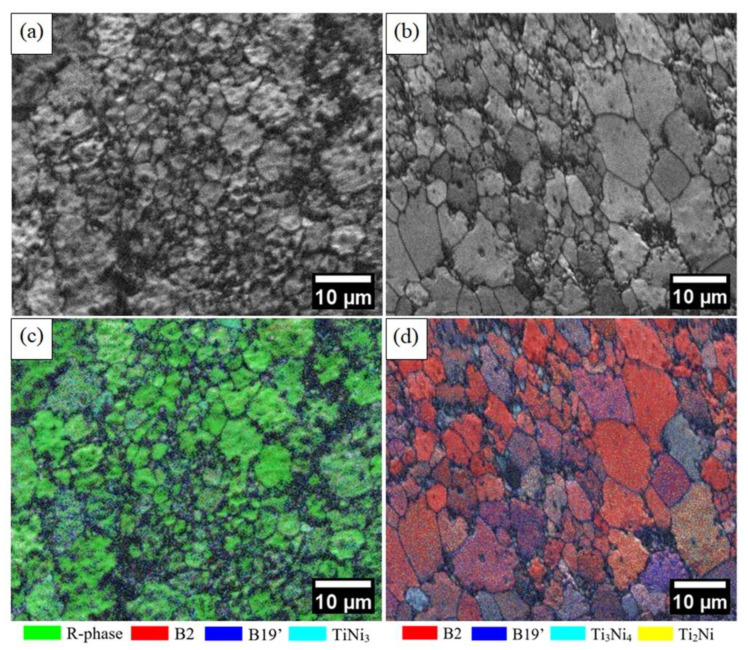
The image quality (IQ) map: (**a**) MW and (**b**) CS sample. EBSD phase maps + IQ map: (**c**) MW sample with the R-phase in green, B2 in red, B19′ in blue, and TiNi_3_ in cyan; (**d**) CS sample with B2 in red, B19′ in blue, Ti_3_Ni_4_ in cyan, and Ti_2_Ni in yellow.

**Figure 7 materials-15-05506-f007:**
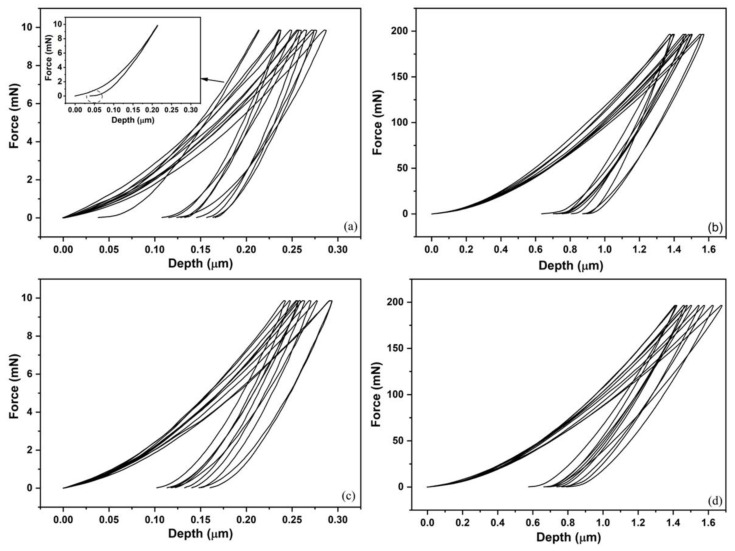
Force vs. depth (loading and unloading curves). (**a**,**b**) MW sample and (**c**,**d**) CS sample; maximum forces of 1.0 gf/9.81 mN and 20.0 gf/196.1 mN, respectively.

**Figure 8 materials-15-05506-f008:**
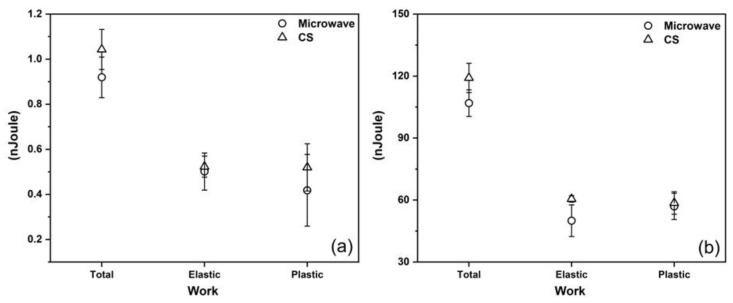
Total, elastic, and plastic works: maximum forces of (**a**) 1.0 gf/9.81 mN and (**b**) 20.0 gf/196.1 mN.

**Table 1 materials-15-05506-t001:** Density, porosity, and pore size values of the porous NiTi alloys prepared by MW and CS, and from the literature results.

Sample	Density (g/cm^3^)	Porosity (%)	Pore Size (µm)
MW	3.31 ± 0.02	51.3 ± 0.29	120 ± 13.84
CS	3.76 ± 0.06	58.2 ± 0.98	155 ± 13.01
* −200 mesh	―	51.9 ± 1.21	97 ± 3.51
* −120 mesh	―	51.5 ± 1.08	149 ± 7.30
* −60 mesh	―	50.1 ± 0.95	238 ± 7.10
* −45 mesh	―	52.9 ± 1.13	294 ± 9.95
** 0 wt.%	5.10 ± 0.20	22.0 ± 0.31	≈26 ± 2.76
** 10 wt.%	3.64 ± 0.02	42.0 ± 0.59	≈120 ± 5.52
** 20 wt.%	3.20 ± 0.02	51.0 ± 0.59	≈151 ± 5.10
** 30 wt.%	2.40 ± 0.60	64.0 ± 1.18	≈178 ± 2.50

*, ** Results from [16] and [37], respectively.

**Table 2 materials-15-05506-t002:** Phase transformation temperatures in degrees Celsius and enthalpy of MW and CS samples. “-” Not detected; “*” undefined.

Sample	Cooling								
	B2 → R			B2 → B19′			R → B19′		
	R_s_	R_p_	R_f_	M′_s_	M′_p_	M′_f_	M_s_	M_p_	M_f_
CS	25.6	20.7	17.8	-	-	-	11.7	7.4	2.9
MW	24.9	19.9	17.7	-	-	-	11.5	7.4	3.4
**Sample**	**Heating**								
	**B19′** **→ R**			**B19′** **→ B2**			**R** **→ B2**		
	**R′_s_**	**R′_p_**	**R′_f_**	**A′_s_**	**A′_p_**	**A′_f_**	**A_s_**	**A_p_**	**A_f_**
CS	25.8	*	*	*	*	*	*	*	58.3
MW	27.4	*	*	*	*	*	*	*	57.6
	**Enthalpy (mJ/g)**								
**Sample**	**First Peak on Cooling**		**Second Peak on Cooling**		**First Peak on Heating**		**Second Peak on Heating**		
CS	1.329		0.856		11.891				
MW	0.279		0.769		11.164				

* It was not possible to accurately determine the transformation temperatures due to overlapping peaks.

## Data Availability

Not applicable.

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
