# Peer review of "Microwave versus Conventional Sintering of NiTi Alloys Processed by Mechanical Alloying"

_materials, 2022, doi:10.3390/ma15165506_

Round 1
Reviewer 1 Report
The present work reports on the synthesis of NiTi porous specimens by microwave and conventional sintering of mechanically alloyed Ni and Ti powders. This work is original and provides some interesting results. It can be accepted for publication in Materials after some revisions.
1) Ni and Ti phases should be pointed out in the milled powders as shown in Fig. 3c.
2) The formation mechanism of laminated structure after mechanical alloying of Ti and Ni powder is required.
3) Why were the conventional sintered samples quenched in water. Quenching may influence the properties of such samples.
Author Response
Dear Reviewer,
we would like to acknowledge the valuable comments performed by the editor that kindly accepted to revise our manuscript as well the who have critically reviewed our article and suggested some appropriate corrections, which helped to improve the paper. All the comments were answered and checked carefully in the manuscript. We expected that modifications introduced in the manuscript are clear and concise enough as required by the Reviewers.

Reviewer 2 Report
In this paper comparison between two sintering processes, microwave and conventional sintering, for the manufacture of NiTi porous specimens are presented
The structure of the paper does not fulfills the structure of a research article.
Five keywords (expressions) are included by the authors.
The Introduction section provide sufficient background information for readers in the immediate field to understand the problem that this study addresses.
o In the Results section, the authors present and interpret the results of the performed experiments.
o The paper ends with the Conclusions part. In this section the authors mention the conclusions of their research study.
The studies are not recently developed, SEM images being from 2019 !
I suggest to Reconsider after Minor Revisions for the following reasons:
1. Authors should highlight the novelty of the present work
2. Moderate English changes required
3. The scale of the images from Figure 4. and Figure 5 are hard visible and are not at the same values
4 What is the milling temperature effect on the samples properties?
5. Is missing the section with number 4, possible typo error.
6. What is the atomic concentration of investigated samples?
Author Response

(The authors gave the same response as above.)

Reviewer 3 Report
In the manuscript, the authors have attempted to manufacture SMA using 2 different routes and then compared the performance of both these methods. (i.e. MW and CS) the manuscript as well written and has the potential to get published. However, a few modifications below can further enhance the quality of the manuscript.
Introduction section:
· From line no. 56 to 77, claims are made with little explanation. These separate small paragraphs shall be combined to make a single paragraph and clearly mention the state of art.
· Lines 78 to 86: Authors have mentioned that Microwave sintering is compared with conventional sintering as well as the characterization part. Can the authors elaborate the same in a proper manner? It seems confusing in the current state, what is the objective of the study?
· Research gap needs to be pointed out clearly
· Introduction section needs to highlight the effect of grain size and pore size and final performance and what the studies have reported till date.
Experimental work section:
· A diagram or flow chart for the process MW and CS is required for better clarity of the readers.
Results and discussion work section:
· Figure 2 overlap of DSC curves are mentioned. What is the reason for the same? Authors needs to clearly mention and rephrase the same.
· Some phases in MW and CS are not detected / undefined. What can be the probable reason for the same? Any alternative suggestions to infer the phases?
· Shape memory effect has not been experimented or analysed. A section about the same is required as it is the prime most property needed from the alloy.
· Mechanical testing is missing. A section on the proposed values or strength shall be mentioned as the alloy needs mechanical properties also.
Conclusions:
· Authors have claimed that MW produced parts have closer density to human bone. The claim is difficult to make without proper bio-medical testing. Can authors provide proper justification for same?
Grammar check, spell check and punctuation also needs to be checked thorough out the entire manuscript.
Author Response

(The authors gave the same response as above.)

Round 2
Reviewer 3 Report
The manuscript has significant changes. It can now be accepted.